# The PKG Inhibitor CN238 Affords Functional Protection of Photoreceptors and Ganglion Cells against Retinal Degeneration

**DOI:** 10.3390/ijms242015277

**Published:** 2023-10-17

**Authors:** Arianna Tolone, Wadood Haq, Alexandra Fachinger, Akanksha Roy, Sandeep Kesh, Andreas Rentsch, Sophie Wucherpfennig, Yu Zhu, John Groten, Frank Schwede, Tushar Tomar, Friedrich W. Herberg, Vasilica Nache, François Paquet-Durand

**Affiliations:** 1Cell Death Mechanism Group, Institute for Ophthalmic Research, Eberhard-Karls-Universität Tübingen, 72076 Tübingen, Germany; arianna.tolone@live.it (A.T.); yu.zhu@uni-tuebingen.de (Y.Z.); 2Neuroretinal Electrophysiology and Imaging, Institute for Ophthalmic Research, Eberhard-Karls-Universität Tübingen, 72076 Tübingen, Germany; wadood.haq@uni-tuebingen.de; 3Biochemistry Department, University of Kassel, 34132 Kassel, Germany; alexfachinger@uni-kassel.de (A.F.); herberg@uni-kassel.de (F.W.H.); 4PamGene International B.V., 5211 ‘s-Hertogenbosch, The Netherlands; akanksha.roy.in@gmail.com (A.R.); jgroten@pamgene.com (J.G.); ttomar@pamgene.com (T.T.); 5Institute of Physiology II, University Hospital Jena, Friedrich Schiller University Jena, 07743 Jena, Germany; sandeep.kesh@med.uni-jena.de (S.K.); sop.wucherpfennig@yahoo.de (S.W.); vasilica.nache@med.uni-jena.de (V.N.); 6Biolog Life Science Institute GmbH & Co. KG, 28199 Bremen, Germany; ar@biolog.de (A.R.); fs@biolog.de (F.S.)

**Keywords:** neuroprotection, retinitis pigmentosa, cGMP, PKG or PKG1, Kv1.3, Kv1.6, functional vision preservation, ERG, cGMP analogues or PKG inhibitors

## Abstract

Hereditary retinal degeneration (RD) is often associated with excessive cGMP signalling in photoreceptors. Previous research has shown that inhibition of cGMP-dependent protein kinase G (PKG) can reduce photoreceptor loss in two different RD animal models. In this study, we identified a PKG inhibitor, the cGMP analogue CN238, which preserved photoreceptor viability and functionality in *rd1* and *rd10* mutant mice. Surprisingly, in explanted retinae, CN238 also protected retinal ganglion cells from axotomy-induced retrograde degeneration and preserved their functionality. Furthermore, kinase activity-dependent protein phosphorylation of the PKG target Kv1.6 was reduced in CN238-treated *rd10* retinal explants. Ca^2+^-imaging on *rd10* acute retinal explants revealed delayed retinal ganglion cell repolarization with CN238 treatment, suggesting a PKG-dependent modulation of Kv1-channels. Together, these results highlight the strong neuroprotective capacity of PKG inhibitors for both photoreceptors and retinal ganglion cells, illustrating their broad potential for the treatment of retinal diseases and possibly neurodegenerative diseases in general.

## 1. Introduction

Photoreceptor degeneration is a hallmark of retinal degenerative diseases (RD), which are characterized by primary dysfunction and the degeneration of photoreceptor cells, leading to visual loss and eventually blindness [1]. Because of the genetic heterogeneity of RD-type diseases, no effective therapies are currently available [2]. While the mechanisms causing photoreceptor degeneration are far from being fully understood, high levels of cyclic guanosine monophosphate (cGMP) are known to trigger non-apoptotic photoreceptor cell death in many RD disease models [3,4]. Two commonly used models for studying RD are *rd1* and *rd10* mice characterized by a nonsense (*rd1*) and a missense (*rd10*) mutation in the gene encoding the β-subunit of rod phosphodiesterase-6 (PDE6) [5]. A lack of PDE6 activity leads to cGMP accumulation in photoreceptors [6,7], and emerging RD neuroprotection strategies include targeting pathways downstream of cGMP [3,4].

The prototypic cellular target of cGMP is protein kinase G (PKG), a homodimeric serine/threonine kinase with each subunit harbouring an N-terminal dimerization domain, an auto-inhibitory sequence, two cGMP-binding sites, and a C-terminal kinase domain. The binding of cGMP to PKG releases the auto-inhibitory sequence from the C-terminal catalytic domain, activating PKG [8]. When over-activated, PKG is likely to play a key role in triggering cell death [4,9,10,11,12,13]. In the retina, exceedingly high cGMP levels and strong PKG activation were causally linked to photoreceptor cell death [6,7,14], highlighting PKG as a target for the treatment of RD-type diseases.

To develop new drugs for the treatment of RD, cGMP analogues designed to inhibit PKG have been synthetized [15,16]. These compounds bear an Rp-configured phosphorothioate, which antagonizes the activation of PKG by binding to the cGMP-binding sites, without liberating the catalytic domain [17]. A previous study found that in vivo treatment with the cGMP analogue CN003 rescued photoreceptor viability and function in the genetically distinct *rd1*, *rd2*, and *rd10* animal models [16]. Since then, promising 2nd-generation cGMP analogues have been developed, but their biochemical properties and neuroprotective efficacy have not been examined thus far.

Here, we identified a novel cGMP analogue with strong photoreceptor-protective effects in *rd1* and *rd10* organotypic retinal explant cultures. In long-term *rd10* and wild-type (WT) retinal cultures, studied using micro-electrode arrays (MEAs), this compound preserved photoreceptor function and, surprisingly, also improved retinal ganglion cell (RGC) activity. A combination of multiplex peptide microarray analysis for kinome activity, MEA- and patch-clamp recordings, and Ca^2+^-imaging experiments identified outward rectifying K^+^-channels Kv1.3 and Kv1.6 as possible mediators of PKG-dependent cell death. Given the important functions of Kv1 channels for neuronal repolarization, these data suggest PKG inhibitors as novel neuroprotective drugs with possible applications beyond the retina.

## 2. Results

### 2.1. Novel PKG Inhibitors Protect rd1 Photoreceptors

We selected cyclic nucleotide (CN) analogues of cGMP (Appendix A) that shared structural similarities with the retinoprotective reference compounds CN003 and CN004 [16]. Like the reference compounds, the selected compounds CN007, CN226, and CN238, have a cGMPS backbone containing a sulphur-modified phosphate in Rp configuration, which confers PKG inhibitory properties [17]. The PKG activators CN008 and CN052 were used as an additional reference (Appendix A).

We initially tested the inhibitory Rp-cGMPS analogues on organotypic retinal explant cultures derived from *rd1* mice. Treatment started at P7 and ended at P11, i.e., at the onset of manifest of retinal degeneration in this mouse model [7], and thus a time-point well suited for establishing possible protective effects. As readout for treatment effect, we detected dying cells using TUNEL assays on sections obtained from treated and non-treated (NT) specimens (Figure 1A). A reduction in the percentage of TUNEL-positive cells in the outer nuclear layer (ONL) was taken to indicate decreased photoreceptor degeneration. When compared to *rd1* NT (100%), retinae treated with CN007, CN226, and CN238 (NT = 7.03 ± 1.61; CN007 = 5.04 ± 1.26; CN226 = 6.18 ± 0.66; CN238 = 4.39 ± 0.50) showed a reduction in TUNEL positive cells of ≈32%, ≈17%, and ≈40%, respectively. CN003 (4.79 ± 0.50) and CN004 (4.49 ± 0.45) confirmed their previously seen protective effects [16] (Figure 1B).

We then tested the inhibitory potential of CN238, the most promising compound as assessed by the TUNEL data, on recombinant *human* PKG isoforms PKG1α, PKG1β, and PKG2 (Figure 1C). As a reference, we included CN003 and CN226, with the latter serving as the “negative control”, since CN226 had not shown protection in *rd1* retinal explants. Compared to CN003, the in vitro kinase activity assay indicated an increased potency of CN238 towards PKG1β and revealed CN238 to have partial agonistic effects on PKG1α and PKG2 at high concentrations. Thus, CN238 may modulate or dampen PKG1α and PKG2 activity rather than completely blocking it. On the other hand, CN226 was only weakly inhibiting PKG1β and PKG2, and, in fact, activated PKG1α, in line with our results on *rd1* retinal explants (cf. Figure 1).

### 2.2. PKG Inhibition Preserves Viability and Function of Photoreceptors in rd10 Retinal Explants

We further tested CN238 and CN003 on organotypic retinal explant cultures derived from *rd10* mice. As in *rd1* animals, the *rd10* mutation affects the gene encoding the β-subunit of rod PDE6; however, the *rd10* PDE6 enzyme retains some residual activity, delaying the onset of photoreceptor degeneration until P18 [3,4]. Hence, *rd10* retinae were cultured from P9 to either P17 or P19, or from P12 till P24.

To assess long-term compound effects, we analysed the number of photoreceptor rows (Figure 2A). When the *rd10* retina was treated with CN003 and CN238 until either P17 or P19, there was no statistically significant rescue, likely because of the comparatively late onset of *rd10* degeneration around P18. At P24, however, a highly significant (*p*-value < 0.01) rescue of *rd10* photoreceptors was observed, with an increase in photoreceptor row counts of ≈55% and ≈46% in CN003- and CN238-treated samples (NT = 3.86 ± 1.01; CN003 = 6.01 ± 0.87; CN238 = 5.64 ± 0.32), respectively (Figure 2B). To assess how the treatment with CN238 might have affected rod and cone photoreceptors, a rhodopsin and a cone-arrestin staining was performed. The rhodopsin staining labelled rod outer segments, with a tendency for a stronger label in CN238-treated specimens. Importantly, the cone-arrestin staining and the number of cones per 100 µm of retinal circumference showed a marked and significant cone preservation in CN238-treated specimens (NT = 0.30 ± 0.33; CN238 = 6.81 ± 3.14; n = 3, *p* = 0.022; Appendix A).

To determine whether the increased photoreceptor survival seen with PKG inhibition also translated into improved retinal function, we performed electrophysiological recordings of *rd10* retinal explants employing a micro-electrode array (MEA) system and white-light LED stimulator to apply 500 ms full-field light flashes. This allowed for a selective assessment of photoreceptor functionality by recording light-elicited micro-electroretinograms (µERG) [18]. In this analysis, we included CN238 and CN003 (cf. Figure 1B), as well as CN226, which served as additional “negative” control.

As shown in representative µERG traces (Figure 2C), after 12 days of in vitro culture CN003 and CN238 strongly increased the amplitudes of light-induced retinal responses, when compared to NT and CN226. In particular, the initial negative deflection of the µERG (Figure 2C, arrow) indicated a light-induced hyperpolarization of photoreceptors (i.e., a-wave in conventional ERG) and, thus, their functional preservation with drug treatment. For each retina, recordings were obtained from two different areas located dorsally and ventrally from the centre and averaged (each recording area: 340 × 280 µm; 59 electrodes at 40 µm spacing). Since each MEA electrode captured the integrated signal of multiple photoreceptors within an electrode’s recording range, the total span of the MEA recording field allowed the estimation of the overall light sensitivity of a given retinal explant. We considered a light-induced negative µERG deflection ≥ 1.75-fold above the respective average control baseline to indicate light responsiveness and drew activity maps for MEA electrodes showing light responses (Figure 2D). We then expressed the number of electrodes showing such a response as a percentage of the total (Figure 2E). This analysis revealed that NT retina, and CN226-treated retina, displayed almost no response to light (NT = 3.3% ± 3.9; CN226 = 9.5% ± 11.2), while CN003 (56.4% ± 13.5) and CN238 (81.7% ± 15.7) strongly increased the light responsiveness of treated retinal explants (Figure 2E). In other words, treatment with CN003 or CN238 dramatically increased the retinal area responding to light by ≈17- or ≈25 times, respectively, when compared to NT. Compared to CN003, CN238 increased retinal responsiveness by yet another ≈ 45%, suggesting a further improvement in photoreceptor function and/or the density of the photoreceptors activating a given MEA electrode. This result may evidence non-linear threshold effects where a numerically small improvement in photoreceptor viability may lead to a strong increase in retinal function.

As an additional measure of photoreceptor functionality, we quantified the amplitudes of the initial negative µERG deflection, to rate light-induced photoreceptor hyperpolarization. Compared to NT, CN003- or CN238-treated retinal explants on average displayed a stronger hyperpolarization response to light (NT: −13.6 µV ± 4.6; CN003: −42.1 µV ± 18.0; CN238: −41.5 µV ± 16.1). In contrast, CN226 treatment had only a minor effect on response amplitudes (−22.7 µV ± 14.3) and/or the density of photoreceptors activating a given MEA electrode (Figure 2F).

Taken together, we have established that CN003 and CN238 preserved not only the viability of photoreceptor cells but also their functionality. In these comparisons, the effects of the novel compound CN238 were at least equal, if not superior, to the previous lead compound CN003.

### 2.3. CN238 Prevents Axotomy-Induced Degeneration of Retinal Ganglion Cells

The MEA recordings of treated *rd10* retinal explants revealed an entirely unexpected feature: during retinal explantation, the optic nerve is transsected, leading to the degeneration of most RGCs within 4–7 days of in vitro culture [19,20]. Accordingly, in NT *rd10* retinal explants cultured for 12 days in vitro, RGC spiking activity was virtually extinguished, and this was also true for CN226-treated explants. Yet, the CN003- and CN238-treated explants showed a remarkable preservation of light-stimulus-correlated RGC spiking activity (Figure 3A).

We assessed the light responsiveness of RGCs across the entire MEA chip surface, from two different central recording areas. To perform this, we considered the percentage of MEA electrodes detecting light-related spike activity. Light-induced RGC spike responses were considered to be stimulus-correlated, if post-stimulus activity (600 ms) exceeded the average pre-stimulus activity (500 ms, 100 ms bin). While RGC light responsiveness in the NT and CN226-treated cultures was nearly absent (NT = 3.3% ± 4.9; CN226 = 1.5% ± 1.4), it was strongly and highly significantly increased in CN003- and CN238-treated retinae (CN003 = 53.6% ± 15.5; CN238: 49.2% ± 8.8) (Figure 3B). Compared to NT, the retinal area showing RGC light responses was ≈16- or ≈15-times larger after the CN003 or CN238 treatment, respectively.

The retinal explants used for MEA recordings then underwent a histological workup to assess RGC survival, employing a labelling for RNA-binding protein with multiple splicing (RBPMS), a protein expressed in about 60% of RGCs [21] (Figure 3C). The quantification of RBPMS-positive cells per mm^2^ in the four experimental groups yielded very low RGC counts in the NT and CN226-treated explants (NT = 134.1/mm^2^ ± 117.3; CN226 = 184.1 ± 81.06), while the CN003- and CN238-treated explants displayed significantly larger RGC numbers (CN003 = 514.4 ± 187.7; CN238 = 460.3 ± 153.6). Overall, CN003- and CN238-treated retinal explants showed 3.8- and 3.4-fold higher numbers of RBPMS-positive cells, respectively (Figure 3D).

The magnitude of the rescue effect on axotomized RGCs raised the question of whether this was a direct effect on RGCs, or whether the preservation of *rd10* photoreceptors had indirectly enhanced the survival of *rd10* RGCs. To address this question, we extended our investigation to WT retinal explants, cultured from P12 to P24, treated or not with CN238. The light-induced hyperpolarization response in the µERG obtained from P24 WT explants did not differ between NT and CN238-treated retinae. However, RGC spiking activity was virtually absent in NT, but present after CN238 treatment (Appendix A). The overall photoreceptor activity was obviously higher in WT than in the *rd10* explants; yet, even in the WT situation CN238 treatment improved light responsiveness (NT = 61.7% ± 45.9; CN238 = 98.9% ± 2.6; Appendix A). The light-response amplitude, i.e., the negative deflection of the µERG, was also increased in the CN238-treated explants (NT = −26.2 ± 20.5 µV; CN238 = −79.0 ± 36.9 µV; Appendix A). The most striking effect of CN238 on WT retinae was, however, observed at the level of RGC light-responsiveness: while NT retinae showed nearly no light-correlated activity, CN238 treatment largely preserved RGC light-induced spiking activity across the whole retinal explant (NT = 0.8% ± 2.0; CN238 = 78.3% ± 30.6; Appendix A). In relative terms, CN238 had, thus, increased WT RGC functionality by a striking ≈ 95-times.

Further confirmation came from an investigation of WT RGC viability with RBPMS staining, which showed RGCs survival 2.3-fold higher in CN238-treated retinae (NT = 163.9/mm^2^ ± 96.4; CN238 = 379.3 ± 97.7; Appendix A). In addition, no significant photoreceptor loss was detected in NT and CN238-treated samples (Appendix A). These findings confirmed that CN238 preserved RGC activity despite the axotomy caused by explantation. The WT data also demonstrated that RGC protection was independent of photoreceptor degeneration.

### 2.4. Effects of cGMP-Mediated Inhibition of PKG in Retinal Ganglion Cells

We recently studied the ability of PKG to phosphorylate specific peptides on lysed samples of explant cultures either treated with CN003 or untreated [22]. Compared to WT, several voltage-dependent potassium channels belonging to the Kv1 family (i.e., Kv1.3 (KCNA3), Kv1.6 (KCNA6)) showed increased phosphorylation in *rd1* retinae, and this was significantly reduced in CN003-treated retinae. Since a previous in vivo study found Kv1-type channels to contribute to RGC degeneration after optic nerve axotomy [23], we evaluated possible differences in Kv1 channel phosphorylation in P24 CN238-treated *rd10* retinal explants, using peptide microarray-based Serine/Threonine Kinase (STK) activity assays. This analysis found decreased phosphorylation for approximately 56% of the 142 peptides present on the STK PamChip^®^, out of which 21 peptides showed significantly decreased phosphorylation in *rd10* CN238-treated retinae (Appendix A). Among the proteins that were significantly less phosphorylated was Kv1.6 (KCNA6).

To map the localization of Kv1.3 and Kv1.6 channels in the adult mouse retina, we employed immunostaining for Kv1.3 and Kv1.6 on the retinal cross-sections (Figure 4A) and flat-mounted retinae (Appendix A) of P24 WT mice. This revealed intense Kv1.3 staining in the ONL, inner plexiform layer (IPL), and nerve fibre layer (NFL). Kv1.6 immunoreactivity was detected in the photoreceptor OS, IPL, and NFL. Furthermore, both Kv1.3 and Kv1.6 antibodies co-localized with anti-SMI32, recognizing non-phosphorylated epitopes of neurofilament proteins. The localization of Kv1.3 and Kv1.6 appeared similar on P11 retina WT and *rd1* retina, while in P24 *rd10* retina the then-obvious loss of photoreceptors may have reduced the expression of these channels in the photoreceptor layer (Appendix A).

### 2.5. Down-Stream Effectors of CN238 Include Kv1 Channels

To understand the consequences of CN238- and CN003-induced PKG inhibition on the Kv1-family, we investigated the influence of these cGMP analogues on Kv1.3 channels heterologously expressed in *Xenopus* oocytes by means of a two-electrode voltage clamp. The expression of Kv1.3 channels gave rise to voltage-dependent outward currents with a voltage of half-maximum steady-state activation (*V*_1/2_) of −17.2 ± 1.14 mV (s = 13.1 ± 1.0; *n* = 10; Figure 4B). The plasma membrane incorporation of Kv1.3 channels was confirmed by the strong current block (92.90% ± 2.51, *n* = 8) in the presence of 500 nM margatoxin [24]. Both CN238 and CN003 reduced the maximal Kv1.3 current amplitude in a time-dependent manner by approximately ≈8.1% and ≈29%, respectively (Figure 4C). Moreover, Kv1.3 voltage dependence was not affected by the compounds (Figure 4D,E). When applying the corresponding oxo-compounds CN052 and CN008, which behave as PKG activators [25], we observed no significant increase in Kv1.3-channel activity, likely because the currents were measured at +60 mV, i.e., close to the maximal activation range of the channels.

Potentially, cGMP analogues could also act on cyclic nucleotide–gated ion (CNG) channels, which have also been related to photoreceptor degeneration [26]. Therefore, we examined the effect of CN238 on heterologously expressed CNG channels in *Xenopus* oocytes. Here, the *EC*_50_ for cGMP-dependent currents was 39.5 µM and 10.7 µM for rod and cone isoforms, respectively (Appendix A). In the absence of cGMP, CN238 could not efficiently activate retinal CNG-channels, yet when co-applied with cGMP, it significantly lowered channel activity. The sensitivity of rod and cone CNG-channels for cGMP was reduced by a factor of ≈6 for both channel types. We recently reported a similar inhibitory effect also for CN003 on retinal CNG-channels [27]. Thus, both CN238 and CN003 significantly reduced CNG-channel activation at pathological high cGMP concentrations.

Overall, our results suggest that the lack of PKG-induced phosphorylation of Kv1.3-channels triggered a voltage-independent reduction in channel activity, with the effect of CN003 exceeding that of CN238. Hence, under RD conditions, these PKG inhibitors should indirectly reduce K^+^ outflow through Kv1.3 channels while, at the same time, they may directly reduce Na^+^ and Ca^2+^-influx through CNG-channels.

### 2.6. CN238 Inhibits Kv1-Mediated Ca^2+^ Extrusion

To relate the effect of CN238 to retinal function and the outward current of Kv1, we performed MEA-based µERG recordings on acute retinal explants of WT mice in the presence of CN238, its corresponding PKG-activator CN052, and the selective inhibitor of Kv1.3 and Kv1.6 margatoxin (Mrg; Figure 5A).

The light-induced hyperpolarization in µERG (negative a-wave deflection, Figure 5A,B) of acute retinal explants exposed to Mrg did not differ from control (ctr: −88.76 ± 14.08 and Mrg: −79.59 ± 10.91 µV). In contrast, in explants exposed to CN238 (−16.18 ± 8.6 µV) and CN052 (−39.46 ± 10.73 µV), light-induced hyperpolarization significantly decreased 5.5- and 2.4-fold compared to control (Figure 5B). The light-correlated responses of RGCs in the CN052 groups were almost identical to those in the control groups (CN052: 77.53 ± 40.46 and ctr: 75.14 ± 34.29 spike count), yet they were significantly reduced in the Mrg (4-fold) and CN238 (7-fold) groups (Mrg: 19.02 ± 9.86 and CN238: 10.54 ± 8.51 spike count; Figure 5C), indicating decreased spike generation due to Kv1 antagonization.

To validate the light-induced responses of RGCs (retinal network-mediated RGC responses), we evaluated RGC responses elicited independent of light through direct potassium chloride (KCl) stimulation, using Ca^2+^-imaging on acute retinal explants derived from adult, blind *rd10* mice. The effect of the depolarizing KCl-stimulus was recorded in the presence of either CN238, CN052, Mrg, or with no compound as the control (Figure 5D). The Mrg and CN238 data revealed a significant delay in the clearance of intracellular Ca^2+^ (at 90 s: Mrg 2.04-fold and CN238 1.84-fold higher than control (4.28 ± 0.30); at 180 s: Mrg 4.81-fold and CN238 2.34-fold higher than control (1.26 ± 0.13)), while CN052-treated retinae behaved similar to control (Figure 5E). Together, these results strongly suggested an inhibition of Kv1-mediated RGC repolarization by CN238.

## 3. Discussion

The excessive accumulation of cGMP in photoreceptors has long since been established as a trigger for photoreceptor loss in rare, RD-type diseases [14]. Using the novel inhibitory cGMP-analogue CN238, we validated PKG as a critical effector of cGMP-dependent cell death. Compared to the parent compound CN003, CN238 affords superior functional retinal protection. Unexpectedly, the protective effect of PKG inhibition extended beyond photoreceptors to axotomized RGCs, neurons whose degeneration is underlying common retinal diseases such as glaucoma or diabetic retinopathy. Importantly, a drug-mediated rescue of axotomized RGCs of the magnitude seen in this study has not been reported before. This emphasizes the importance of PKG for neuronal cell death and highlights PKG inhibition as a new therapeutic approach for the treatment of neurodegenerative diseases in general.

### 3.1. cGMP Analogues as PKG Inhibitors

Analogues of cGMP carrying an Rp-configurated phosphorothioate modification were first described as potent and selective PKG inhibitors in the early 1990s [15,28]. While clinically used kinase inhibitors typically block the ATP-binding site present on all kinases [29], cGMP analogues target the cGMP-binding site present only on PKG, thereby affording an extraordinary selectivity for PKG.

With CN238, we identified a PKG inhibitor that preserved photoreceptor viability and function in *rd1* and in *rd10* retina. This novel compound was found to have improved potency when compared to the reference compound CN003, a known PKG inhibitor with protective effects in *rd1*, *rd2*, and *rd10* mice in vivo [16]. Both carry the ß-phenyl-1, N^2^-etheno (PET) group [30] and differ only by the additional methyl group in CN238. Interestingly, the compound CN007, which has a similar N^2^-etheno modification, was moderately photoreceptor protective, while CN226, which lacked such a R_2_-R_3_ modification (Appendix A), did not afford *rd1* photoreceptor protection. This was corroborated by studies on an *rd10* retina, where CN226 showed significantly less preservation of photoreceptor function than CN003 or CN238.

The PET-group enhances the lipophilicity of Rp-modified cGMP analogues, making it easier for these compounds to reach their target site inside the cell. In addition, the PET-group can inhibit cyclic nucleotide gated ion (CNG) channels, albeit with an efficacy that is ≈2 log units lower than for PKG inhibition [30]. The electrophysiological recordings on *Xenopus laevis* oocytes showed that both CN003 and CN238 can reduce CNG- and Kv1.3-channel activity. This may suggest that the protective effects of CN003 and CN238 on photoreceptors could stem from combined PKG- and CNG-channel inhibition. Although CNG-channel activity was for many years considered to be a driver of photoreceptor degeneration [26,31], numerous studies in the last two decades have explored the use of CNG-channel blockers, essentially without tangible results. Moreover, a recent study found that the selective block of CNG channels with L-*cis*-diltiazem increased rather than prevented photoreceptor cell death [32]. These conflicting results suggest that CNG-channel inhibition alone may not slow the rate of cell death. Yet, cGMP analogues down-regulating the activity of both PKG and CNG channels may hold promise for RD treatment development.

### 3.2. PKG Inhibition Affords Multilevel Neuronal Protection

The PKG inhibitors CN003 and CN238 robustly preserved *rd10* photoreceptor function as assessed by retinal µERG field-potentials recording using MEA. Each of the 59 MEA electrodes covered an area of approx. 80 µm^2^; thus, the recorded field potentials likely originated from thousands of photoreceptors [33]. Likewise, the amplitude of the initial negative deflection in the µERG represents the sum response of a large photoreceptor number in each recording field. Moreover, the recording of µERGs at different retinal locations allowed the creation of spatial activity maps for light responsiveness, both for treated and untreated tissues. The comparison of such maps in *rd10* retinae revealed large differences between CN003/CN238-treated and untreated specimens, not only in terms of µERG amplitudes but also in terms of the retinal areas showing responses to light flashes. This demonstrates the magnitude of photoreceptor protection over a large retinal area, an effect that was corroborated by the histological examinations and especially the preservation of cone photoreceptors.

The deleterious effects of high cGMP on photoreceptor viability had already been established in the 1970s [6,14]. Yet, the role of PKG as an important mediator of cGMP-dependent photoreceptor cell death was recognized only more recently [4,7]. Our finding that PKG inhibition, in addition to overall photoreceptor protection [16] and cone preservation specifically, also prevented the demise of RGCs in both WT and *rd10* long-term retinal explant cultures was unexpected. The transection of the optic nerve is a massive insult, known to cause rapid RGC function loss and degeneration [20,34]. Surprisingly, our MEA recordings indicated a striking preservation of RGC function, concomitant with a marked and significant increase in morphological RGC survival. A recent MEA study on retinal explant cultures found that in WT retinae, RGC activity gradually decreased to essentially zero within 14 days of culture. More importantly, RGC responses to light stimulation were no longer observed beyond 7 days of culture [19], a result that corresponds to our observations on both WT and *rd10* retinae. Yet, in treated WT retinae, the MEA recordings revealed strong light-correlated RGC responses, and the improved RGC survival after CN238 treatment was confirmed using immunohistochemical analysis with the RGC marker RBPMS [21,35]. RBPMS staining revealed similar RGC numbers in both *rd10* and WT untreated retinal explants, and in WT retinae, photoreceptor row counts did not show differences between the CN238-treated and untreated groups. Hence, in both the WT and the *rd10* situation, long-term retinal explant cultures displayed a loss of RGCs over time, and their rescue by CN238 was independent of photoreceptor survival. The failure of CN226 to preserve RGC viability and function in *rd10* retinal explants indicates that RGC survival may indeed be connected to PKG inhibition.

The strong RGC protection seen with CN003 and CN238 makes them attractive for therapy development beyond photoreceptor diseases. Indeed, RGC degeneration is a hallmark of several retinal diseases, with only limited treatment options to date. This includes glaucoma [36] and diabetic retinopathy [37], as well as exudative and non-exudative age-related macular degeneration [38,39]. How exactly PKG inhibition may afford RGC neuroprotection is unclear at present, although numerous studies have invoked detrimental effects of nitric oxide synthase and nitric oxide (NO) on RGCs [40,41], also in optic nerve injury [42]. Since NO stimulates soluble guanylyl cyclase to produce cGMP and activate PKG [43], increased NO production in injured RGCs will likely also cause the overactivation of PKG, providing a rationale for the use of PKG inhibitors for RGC neuroprotection. The axotomy-induced degeneration of RGCs resembles a Wallerian-like retrograde degeneration [44,45]. The fact that PKG inhibition significantly reduces this type of degeneration could, thus, indicate that PKG inhibitors may be applicable even more broadly in neurodegenerative conditions characterized by axonal damage, such as spinal cord injury or multiple sclerosis.

Further considerations for PKG-inhibition-mediated neuroprotection relate to the fact that PKG activity is known to relax smooth muscles and lower blood pressure [46]. In our present study, we used organotypic retinal explant cultures devoid of retinal or choroidal vasculature. We, therefore, cannot exclude the possibility that the PKG-inhibiting compounds employed could have additional effects on retinal blood vessels. However, we note that the PKG effects on blood pressure appear to be mediated in particular by PKG1α [47]. Since CN238 has a relatively low effect on PKG1α vs. PKG1β and, moreover, exhibits partial agnostic behaviour on PKG1α, but not on PKG1β, this compound is unlikely to significantly impact blood pressure. This property of CN238 may, thus, improve the perspectives for its clinical application.

### 3.3. PKG Inhibition and Neuronal Cell Survival: Significance of Kv1 and CNG Channels

Previous in vivo studies have reported that the inhibition of the voltage-dependent potassium channels Kv1.1 and Kv1.3 rescued RGCs after optic nerve axotomy [23,48,49]. The Kv1 family consists mainly of slow-activating and inactivating delayed rectifier channels [50]. Recently, we identified several members of the Kv1 family (i.e., Kv1.2, Kv1.3, and Kv1.6) as potential substrates of PKG in photoreceptor-like 661W cells [51]. In another study, we measured the ability of PKG to phosphorylate specific substrates in retinal explant samples derived from *rd1* and WT, treated or not with CN003. Among the substrates highly phosphorylated in *rd1* but not in WT were Kv1.3 and Kv1.6 channels, and this phosphorylation was significantly reduced by CN003 [22].

Immunofluorescence on retinal cross-sections and flat-mounts derived from WT P24 mice localized Kv1.3 and Kv1.6 to the ONL and the nerve fibre layer (NFL), consistent with earlier observations on the distribution of potassium channels in the mammalian retina [50,52]. The NFL tract proximal to the RGC soma consists of unmyelinated axons characterized by mitochondria-rich varicosities and a high level of Na^+^/K^+^-ATPase [53], suggesting a high energy demand in this area [54]. Analysis of kinase activity in retinal explant cultures derived from *rd10* P24 mice showed reduced phosphorylation of the Kv1.6 channel in CN238-treated retinae compared to untreated retinae. In addition, the results obtained from both electrophysiological recordings on *Xenopus laevis* oocytes and Ca^2+^-imaging on *rd10* acute retinal explants confirmed that CN238 exerts margatoxin-like effects, likely down-regulating Kv1-channel activity. In fact, the delayed repolarization after treatment with margatoxin and CN238 suggests that the latter inhibits Kv1 channels. Conversely, activation of PKG in RGCs directly or indirectly increases phosphorylation of its targets Kv1.3 and Kv1.6, increasing K^+^ efflux from these channels. The greater the K^+^ efflux from the cell, the greater the ionic imbalance between the inside and outside of the cell, which needs to be compensated by increased Na^+^/K^+^-ATPase activity [55]. However, an overall greater energy expenditure will accelerate the demise of a cell that is already compromised by an axotomy. Similarly, in RD-mutant photoreceptors, a high loss of K^+^ caused by the PKG-dependent activation of Kv1.3 and Kv1.6 will increase Na^+^/K^+^-ATPase activity in the inner segment (IS), likely precipitating cell death (Figure 6). Future studies may determine whether the direct targeting of Kv1 channels in photoreceptors or RGCs can produce protective effects like those mediated by CN003 or CN238.

Moreover, µERG recordings on acute WT retinal explants showed only a minor modulation of photoreceptor light-induced responses with margatoxin, while CN238 and the PKG activator CN052 reduced photoreceptor response amplitudes. This suggests that both compounds reduce CNG-channel activity, likely because of the PET moiety they are carrying [30]. While our data show that CN238 is far more effective on PKG, the EC_50_ values obtained for the CNG channels indicate that this compound can also modulate their activity. Such a reduction in CNG-channel-mediated Na^+^/Ca^2+^ influx would further decrease photoreceptor energy expenditure, likely improving photoreceptor survival.

## 4. Methods

Animals: Experiments involving C3H *Pde6b^rd1^*^/*rd1*^ (*rd1*), congenic C3H *Pde6b*^+/+^ wild-type (C3H), C57BL/6J wild-type (C57) and C57BL/6J *Pde6b^rd10/rd10^* (*rd10*) mice were compliant with the ARVO declaration and the German Federal Government’s animal protection law and approved by the University of Tübingen’s animal welfare office. Experiments with *Xenopus laevis* frogs were approved by the animal ethics committee of the University of Jena and Thüringer Landesamt für Verbraucherschutz.

cGMP analogues synthesis: Cyclic nucleotide analogues used in this study were produced by Biolog Life Science Institute GmbH & Co. KG according to previously described methods [15] (https://patentscope.wipo.int/search/en/detail.jsf?docId=WO2018010965 accessed on 10 October 2023).

In vitro PKG activation/inhibition assay: FLAG-Strep-Strep-tagged human PKG1α (2–671) WT, human PKG1β (4–686) WT, and human PKG2 (1–762) WT, were expressed in HEK293T cells. Cells were transfected at 80% confluency using polyethyleneimine (Polysciences Europe GmbH, Hirschberg an der Bergstraße, Germany). Cells were lysed using 50 mM Tris-HCl (pH 7.3), 150 mM NaCl, 0.5 mM TCEP, 0.4% Tween, protease, and phosphatase-inhibitors (Roche, Mannheim, Germany). For purification, we employed Strep- Tactin^®^ Superflow^®^ resin (IBA GmbH, Goettingen, Germany). Subsequent washing with 366 mM Na_2_HPO_4_, 134 mM NaH_2_PO_4_ (pH 7.3) and 0.5 mM TCEP at room temperature released any residual nucleotides from their respective nucleotide binding pockets. Strep-tagged proteins were eluted with 200 mM Tris-HCl (pH 8), 300 mM NaCl, 2 mM EDTA, and 5 mM desthiobiotin (IBA GmbH, Goettingen, Germany), and stored at 4 °C in 50 mM Tris-HCl buffer (pH 7.3), containing 150 mM NaCl and 0.5 mM TCEP.

PKG kinase activity was assayed using a coupled spectrophotometric assay [56] in a clear 384 well PS-microplate (Greiner Bio-One, Frickenhausen, Germany) in a CLARIOstar plate reader (BMG LABTECH, Ortenberg, Germany). The assay mixture contained 100 mM MOPS (pH 7.0), 10 mM MgCl2, 1 mM ATP, 1 mM phosphoenolpyruvate, 15.1 U/mL lactate dehydrogenase, 8.4 U/mL pyruvate kinase, 230 µM reduced nicotinamide adenine dinucleotide, 0.1 mg/mL BSA, 5 mM β-mercaptoethanol and, as PKG substrate, 1 mM VASPtide (RRKVSKQE; GeneCust, Boynes, France). PKG activation was determined using cGMP (Appendix A) and the cGMP analogues CN003, CN226, and CN238 in dilution series ranging from 100 µM to 5.1 nM. Inhibition studies were performed by adding each PKG isoform supplemented with 2 µM cGMP to the assay mix and the respective cGMP analogue in dilutions ranging from 100 µM to 5.1 nM.

Heterologous expression of retinal CNG-channels in *Xenopus laevis* oocytes: Oocytes were extracted from South African *Xenopus laevis* female frogs under anaesthesia with 0.1% tricaine (pH = 7.1; MS-222, Parmaq, Hampshire, UK). After incubation in Ca^2+^-free solution (in mM: 82.5 NaCl, 2 KCl, 1 MgCl_2_, and 5 Hepes, pH 7.4) containing 3 mg/mL collagenase A (Roche Diagnostics, Mannheim, Germany) for 105 min, oocytes of stage IV and V were selected and defolliculated. Bovine CNGA1 (NM_174278.2) and CNGB1a subunits (NM_181019.2) from rods, human CNGA3 (NM_001298.2) and CNGB3 subunits (NM_019098.4) from cones and rat Kv1.3 channels (NM_X16001.1) were subcloned into the pGEMHE vector for RNA synthesis [57]. Oocytes were injected with ~40–70 nL cRNA, encoding for the respective ion channels. The ratio of CNGA3 to CNGB3 cRNA was 1:2.5 [58] and of CNGA1 to CNGB1a cRNA was 1:4 [59,60]. Oocytes were kept at 18 °C for 2 to 7 days in Barth’s medium to allow for protein synthesis and trafficking to the plasma membrane. The oocyte vitelline membrane was removed prior to patch-clamp experiments. To confirm the presence of both subunit types within the CNG-channel structure, control experiments were performed as previously published [27,32]. Additionally, oocytes from Ecocyte Bioscience (Castrop-Rauxel, Dortmund, Germany) were used.

Electrophysiology on *Xenopus laevis* oocytes. Whole-cell Kv1.3 currents from Xenopus oocytes were recorded with the two-electrode voltage clamp technique (725C amplifier, Warner Instruments, LLC, Hampden, MA, USA) at room temperature. Glass electrodes (Brand GmbH + Co KG, Wertheim, Germany) were filled with 3 M KCl solution (resistance: 0.3–1.0 MΩ). The bath solution contained 96 mM NaCl, 2 mM KCl, 1.8 mM CaCl_2_, 1 mM MgCl_2_ and 10 mM HEPES, pH = 7.4. Currents were triggered by test potentials from −100 to +90 mV in 10 mV increments from a holding potential of −90 mV. The duration of test potentials was 650 ms and their frequency was 5.00 kHz, while the sampling rate was 200 µs. Cells were held at −90 mV for 20 sec between pulse cycles to avoid cumulative inactivation. For data acquisition, we employed Patchmaster software (v2*90.5 09, HEKA Elektronik GmbH, Reutlingen, Germany). Effects of cGMP analogues (50 µM) were determined from the peak current during the test potential at +60 mV. The steady-state activation was determined from the amplitude of the instantaneous tail currents at 0 mV, which was the voltage pulse, with a duration of 200 ms, following the depolarizing test potentials [61]. For quantifying compound effects on channel voltage dependence, we used the relative tail currents, I, which were determined from the amplitude ratio of the instantaneous current at the actual potential with respect to the maximal instantaneous current, recorded at maximal depolarization. The experimental data points measured at different voltages fitted with Boltzmann’s equation:(1)$$\frac{I}{I_{max}}=\frac{1}{1+e^{\frac{\left(V_{1/2}-V_{m}\right)}{s}}}$$where V_1/2_ is the midpoint voltage of half-maximum activation, V_m_ is the membrane potential, s is the slope factor, and I_max_ is the maximum current.

CNG-channel activity was investigated using patch-clamp recordings performed on inside-out patches from *Xenopus laevis* oocytes, with an Axopatch 200B patch-clamp amplifier (Axon Instruments, Foster City, CA, USA) and the PatchMaster software (v2*90.5 09, HEKA Elektronik GmbH, Reutlingen, Germany). The sampling rate was 5 kHz and the filter implemented in the amplifier was set to 2 kHz. CNG-channel currents were recorded at −100 and +100 mV, from a holding potential of 0 mV. For the analysis, only a time interval at +100 mV was used, where the current was measured under steady-state conditions. Bath and patch-pipettes solution contained 140 mM NaCl, 5 mM KCl, 1 mM EGTA, and 10 mM HEPES (pH 7.4). Ligand concentrations were verified prior to the experiments using UV-spectroscopy (Thermo NanoDrop 2000c, Thermo Fisher Scientific GmbH, Bremen, Germany). Patch pipettes were pulled from borosilicate glass tubing (outer diameter 2.0 mm, inner diameter 1.0 mm, resistance: 0.7–1.7 MΩ; Hilgenberg GmbH, Malsfeld, Germany).

For concentration–activation relationships, rod and cone CNG-channels were first exposed to a cGMP-free solution, then to intermediate concentrations of cGMP, and finally to a saturating concentration of 3 mM. The current in the absence of cGMP was subtracted from the cGMP-induced current. The respective experimental data points fitted with the Hill equation:(2)$$\frac{I}{I_{max}}=\frac{1}{1+{\left(\frac{{EC}_{50}}{x}\right)}^{H}}$$
where *I* is the current amplitude, *I_max_* is the maximum current induced by a saturating ligand concentration, *x* is the ligand concentration, *EC*_50_ is the ligand concentration of half maximum effect, and *H* is the Hill coefficient. The effects of cGMP analogues on CNG channels were determined by comparing the cGMP-induced current amplitudes obtained in their presence (50 µM) and absence. Data obtained were analysed using Fitmaster software (v2*90.5 09, HEKA Elektronik GmbH, Reutlingen, Germany).

Organotypic retinal explant cultures. Retinal explants derived from C57, *rd10*, *rd1*, and C3H animals were prepared as previously described [62]. Post-natal day (P)5 *rd1*, P9 or P12 *rd10* animals were sacrificed, and their eyes enucleated and incubated in R16 retinal culture medium (074-91252A, Gibco, Paisley, Scotland) with 0.12% proteinase K (ICN Biomedicals Inc., Costa Mesa, CA, USA) for 15 min at 37 °C. Proteinase K activity was blocked by the addition of 20% foetal bovine serum (FBS) (F7524, Sigma, Hamburg, Germany). The anterior segment, lens, vitreous, sclera, and choroid were removed, while the RPE remained attached to the retina. Explants were cut into a four-wedged shape, transferred to a culture membrane insert in a six-well plate (Corning Life Sciences, Corning, NY, USA) with the RPE facing the membrane and incubated with complete R16 medium with supplements, but free of serum and antibiotics [62], in a humidified incubator (5% CO_2_) at 37 °C. After 48h, the retinae were either exposed to 50 µM of different cGMP analogues (dissolved in water) or kept as an untreated control. The medium was changed every second day. The culturing paradigm was from P5 to P11 for *rd1*, and from P9 to either P17/P19, or from P12 to P24 for *rd10*. Culturing was stopped by 45 min fixation in 4% paraformaldehyde (PFA), cryoprotected with graded sucrose solutions and embedded in Tissue-Tek O.C.T. compound (Sakura Finetek Europe, Alphen aan den Rijn, Netherlands). To allow an assessment of retinal explant culture quality, a comparison between the in vivo (i.e., acute preparation) and in vitro (i.e., after 6–12 days of culture) situation is presented in Appendix A.

Histology: For retinal cross-section preparation, eyes were marked nasally, and the cornea, iris, lens, and vitreous were removed. Eyecups were fixed and cryoprotected as above for retinal cultures. Tissue sections of 12-14 µm were prepared using Thermo Scientific NX50 microtome (Thermo Scientific, Waltham, MA, USA) and thaw-mounted onto Superfrost Plus glass slides (R. Langenbrinck, Emmendingen, Germany). For retinal flat mount preparations, eyes were enucleated and the anterior segment, lens, vitreous, sclera, choroids and RPE were removed. Retinae were fixed with 4% PFA and directly prepared for immunostaining.

TUNEL assay: cGMP analogues were tested for their effect on photoreceptor cell death, using a terminal deoxynucleotidyl transferase dUTP nick end labelling (TUNEL) assay [63] (Sigma-Aldrich in situ Cell Death Detection Kit, 11684795910, red fluorescence). 4′,6-diamidino-2-phenylindole (DAPI) contained in the mounting medium (Vectashield with DAPI; Vector Laboratories, Burlingame, CA, USA) was used as nuclear counterstain.

Immunofluorescence: Immunostaining with primary antibody against rabbit RBPMS (1:500; Abcam, #ab194213, Cambridge, UK), rabbit rhodopsin (1:300, Sigma-Aldrich, #ab5316, St. Louis, MI, USA) and rabbit cone arrestin (1:200, Sigma Aldrich, #ab15282) was performed on 12 µm thick retinal explant cryosections by incubating at 4 °C overnight. Immunostaining with primary antibody against rabbit KCNA3 (1:200; Alomone Labs, #APC-101, Jerusalem, Israel), rabbit KCNA6 (1:300; Alomone Labs, #APC-003) and mouse SMI32 (1:1000; Biolegend, #801707, San Diego, CA, USA) was performed on 14 µm thick retinal cross-sections and retinal flat mounts by incubating at 4 °C overnight. Secondary antibodies were conjugated with Alexa Fluor 488 or 568 (Thermo Fisher Scientific, Sindelfingen, Germany). Sections were mounted with Vectashield medium containing DAPI (Vector Laboratories Newark, CA, USA).

Microscopy and image processing: Images were captured using 7 Z-stacks with maximum intensity projection (MIP) on a Zeiss Axio Imager Z1 ApoTome Microscope MRm digital camera (Zeiss, Oberkochen, Germany) with a 20x APOCHROMAT objective. For details on the filter sets for the fluorophores used, see Appendix A. For the positively labelled cell quantification, pictures were captured on at least six different areas of the retinal explant, from at least four different animals per genotype. Adobe Photoshop (CS5Adobe Systems Incorporated, San Jose, CA, USA) was used for image processing.

Ex vivo retinal function test. Prior to recording, organotypic retinal cultures were kept in the dark for at least 12 h. Further manipulations were performed under dim red light. Retinae were divided into two equal halves, of which one was placed on the recording chamber’s electrode array and kept in the dark. The second retinal half was used for histological analysis (see above). Two recordings were obtained from positions within the retina’s central half. Functional tests were performed in drug-free R16 medium at 37 °C. In contrast, experiments with acute retinal explants were carried out in ACSF medium [64] and exposed either to 50 µM of CN238 or CN052, 50 nM Margatoxin (Alomone Labs, #STM-325), or kept untreated as a control. Electrophysiological recordings and raw data acquisition was as follows:

(A) Micro-electrode array system (MEA). To record light-evoked retinal responses, a micro-electrode array system (MEA; USB-MEA60-Up-BC-System-E, Multi-Channel Systems; MCS, Reutlingen, Germany), equipped with HexaMEA 40/10iR-ITO-pr (60 electrodes = 59 recording and one reference electrode) was employed. Recordings were performed at a 25.000 Hz sampling rate to collect unfiltered raw data. The trigger synchronized operation of the light stimulation (LEDD1B T-Cube, Thorlabs, Bergkirchen, Germany) and MEA recording were controlled by a dedicated protocol implemented within the MC-Rack software (v4.6.2, MCS) and the digital I/O—box (MCS). Light stimulation (white light LED, 2350 mW, #MCWHD3, Thorlabs, Bergkirchen, Germany), guided by fibre-optic and optics, was applied from beneath the transparent glass MEA: five full field flashes of 500 ms duration with 20 s intervals. A spectrometer USB4000-UV-VIS-ES (Ocean Optics, Ostfildern, Germany) calibrated the intensity of the applied light stimulation (1.33 × 10 ^14^ photons/cm^2^/s). Electrophysiology data were analysed using custom-developed scripts (MATLAB, The MathWorks, Natick, MA, USA), unless otherwise indicated. MEA-recording files were filtered employing the Butterworth 2nd-order filter (MC-Rack, MC) to extract retinal ganglion cell spikes (high pass 200 Hz) and field potentials (bandpass 2–40 Hz). MEA field potentials are called micro-electroretinograms (µERG) and correspond to the human electroretinogram (ERG), as described by [33]. Filtered data were converted to *.hdf files using MC DataManager (v1.6.1.0). Further data processing was performed in MATLAB (spike and field potential detection) as previously described [18].

(B) Ca^2+^-imaging: For Ca^2+^-imaging of retinal ganglion cells, acute retinal explants derived from *rd10* mice were loaded with the Ca^2+^-indicator fluorescent dye OGB1 (Oregon Green 488 BAPTA-1, Thermo Fischer Scientific, Karlsruhe, Germany) [65]. Recordings were performed using an upright fluorescence microscope (Olympus BX50WI, Germany) equipped with a 20× water-immersion objective (LUMPlan FL, 40X/0.80W, ∞/0, Olympus), a polychromator (VisiChrome, Visitron Systems GmbH, Puchheim, Germany), and a CCD camera (RETIGA-R1, 1360 × 1024 pixels, 16 bits). Fluorescence time-stacks of OGB1 were acquired at 10 Hz (λexc = 470; Olympus U-MNU filter set, 30 ms exposure time, 8-pixel binning) using VisiView software (v3.1, Visitron Systems GmbH, Puchheim, Germany). A KCl stimulus [50 mM, 50 µL drop] was applied close to the target using a microinjection syringe pump (MICRO2T SMARTouch, World Precision Instruments, WPI, Sarasota, FL, USA); the perfusion rate was set to 2 mL/min (MCS perfusion system).

Kinase activity profiling using peptide microarrays: *rd10* retinal explant samples treated with CN238 or untreated were lysed for 30 min on ice with M-PER Mammalian Extraction Buffer (Thermo Fischer Scientific) supplemented with protease and phosphatase inhibitor cocktails (Halt Phosphatase Inhibitor Cocktail, Thermo Fischer Scientific, #78420 and Halt Protease Inhibitor Cocktail EDTA free, Thermo Fischer Scientific, #87785), followed by centrifugation (16,000× *g*, 15 min, 4 °C) and freezing at −80 °C. Protein quantification of retinal lysates was performed using Pierce Coomassie Plus (Bradford) assay kit (Thermo Fischer Scientific).

Serine/Threonine Kinase (STK) activity measurement was performed on PamChips^®^ (PamGene International B.V., ‘s-Hertogenbosch, The Netherlands), comprising four arrays, each with 142 Serine/Threonine-containing peptides. Each assay was performed in duplicate with an assay mix comprising 0.25 µg of protein lysate, protein kinase buffer (PamGene), 0.01% BSA, STK primary antibody mix (PamGene), and 400 µM ATP. PamChips^®^ were first placed in PamStation^®^ and blocked with 2% BSA by pumping up and down through the arrays 30 times and then washed with protein kinase buffer. Then, assay mixture containing the active kinases was applied and pumped up and down through the arrays for 60 min to facilitate interaction between active kinases and immobilized peptides. Peptide phosphorylation by kinases was detected by a secondary antibody conjugated with FITC and directed towards the primary antibody STK [66,67]. Images of the arrays were recorded at multiple exposure times and quantified using BioNavigator^®^ software, v6.3.67.0 (PamGene).

## 5. Conclusions

In the past decade, there has been tremendous progress in the development of new forms of therapy for RD, including gene-, molecular- and stem cell-based therapies [68,69], as well as retinal prostheses [70]. Nevertheless, there is still an important unmet medical need for more broadly applicable therapies that may benefit large groups of RD patients. Our work highlights PKG inhibition as an attractive approach for a mutation-independent treatment of RD. Still, for the clinical development of CN238 into a treatment for RD, several further questions need to be addressed: to limit potential systemic side effects, the clinical administration route will likely be a local, intravitreal (IVT) injection, which requires studies into the intraocular pharmacokinetics. While amphiphilic compounds such as CN238 can permeate the retina, the intraocular retention may be too low to allow for a sustained effect. Hence, CN238 may need to be combined with a suitable drug delivery system, such as liposome- or PLGA-based carriers [71,72], to allow for long-term delivery and to increase the IVT injection intervals. Although further in vivo validation is needed, our results demonstrate the involvement of cGMP/PKG signalling in the degenerative process [4] and identify CN238 as a second-generation drug candidate with superior protective effects on retinal function. Finally, the protective effect on axotomized RGCs may open new perspectives for PKG inhibitors in the treatment of common retinal diseases, such as glaucoma. An extensive description of the different statistical tools and methods used to analyse the various datasets is presented in the Appendix A.

## Figures and Tables

**Figure 1 ijms-24-15277-f001:**
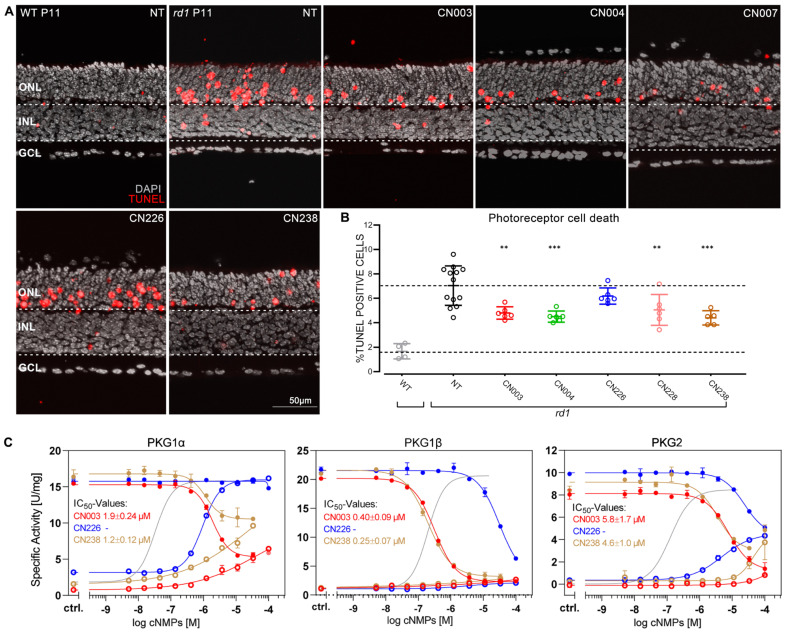
Retinoprotective effects of novel PKG inhibitors in *rd1* retina. (**A**) Post-natal (P) day-11 wild-type (WT) and *rd1* organotypic retinal explant cultures stained with the TUNEL assay (red) showing dying cells, nuclear counterstain with DAPI (grey). *rd1* retinae were treated for four days with 50 µM of different cGMP analogues. (**B**) Quantification of TUNEL-positive cells in the outer nuclear layer (ONL) of sections from A. Cell death rate in NT WT retina shown for comparison. A cell death rate lower than in the non-treated (NT) *rd1* retina was interpreted as evidence for photoreceptor protection. CN003 and CN004 had previously been established as photoreceptor-protective (16) and were used for reference. *n* = 4 to 12 retinae from different animals. Error bars: mean with SD. Statistical analysis: one-way ANOVA followed by Dunnett’s multiple comparisons test; significance levels: ** *p* ≤ 0.01, *** *p* ≤ 0.001. ONL = outer nuclear layer, INL = inner nuclear layer, GCL = ganglion cell layer. (**C**) Agonistic and antagonistic properties of the cGMP analogues CN003, CN226, and CN238. Activation with cGMP was determined for all PKG isoforms (grey solid line, reflecting the activation with unmodified cGMP; cf. Appendix A) utilizing an in vitro kinase assay. Activation and inhibition of 5 nM PKG1α, PKG1β, and PKG2 by CN003, CN226, and CN238. Each analogue was first tested for activation (open circles) and in a second instance kinase inhibition was measured in the presence of 2 μM cGMP (solid circles). IC_50_ values were calculated from dilution series ranging from 100 μM to 5.1 nM obtained in 3 independent measurements.

**Figure 2 ijms-24-15277-f002:**
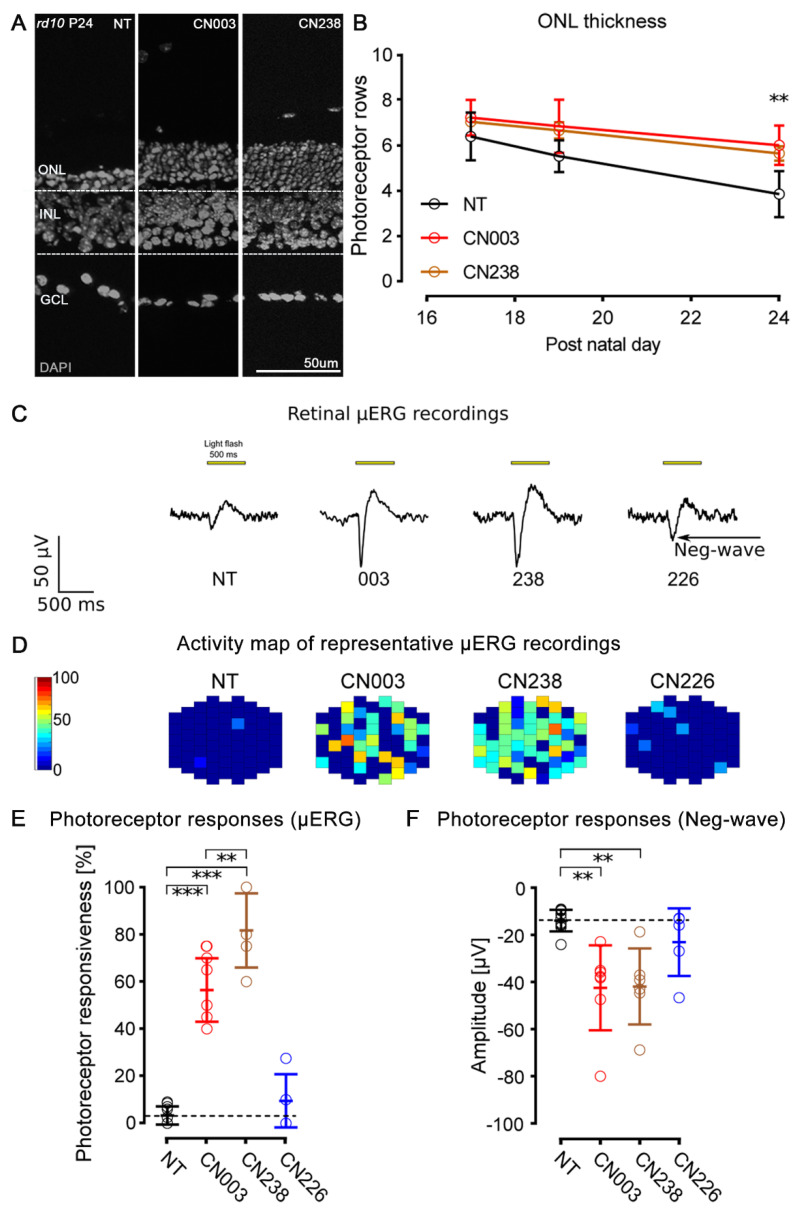
CN238 preserves *rd10* photoreceptor viability and function. *rd10* organotypic retinal explant cultures derived from *rd10* mice treated with cGMP analogues (50 µM). (**A**) Representative sections of post-natal (P) day−24 *rd10* retinal cultures treated or non-treated (NT) and stained with DAPI (grey). (**B**) Photoreceptor row counts in *rd10* retinal explants treated for varying times until either P17, P19, or P24. Row counts above NT were interpreted as evidence for photoreceptor protection. *n* = 5 different retinae from different animals. (**C**) Representative micro-electroretinogram (µERG) traces of NT and treated *rd10* retinal explants. Single-electrode data represent the integrated signal of multiple photoreceptors above a given electrode’s recording field. Light-evoked photoreceptor hyperpolarization is indicated by the initial negative deflection (neg.-wave) of the µERG (arrow). (**D**) Activity map of representative µERG recordings across the 59 electrodes of the micro-electrode-array (MEA), showing light-responses in a spatial context. Each pixel corresponds to a recording electrode with colour-coded µERG response amplitudes. (**E**) Quantification of retinal light responsiveness as a percentage of MEA electrodes displaying µERG negative deflections ≥ 1.75-fold average baseline. *n* = 5 retinae from different animals. (**F**) Average amplitudes of negative deflection in µERG recordings. *n* = 5 retinae from different animals. Error bars: mean with SD. Statistical testing: one-way ANOVA with Dunnett’s multiple comparison test; significance level: ** *p* ≤ 0.01, *** *p* ≤ 0.001. ONL = outer nuclear layer, INL = inner nuclear layer, GCL = ganglion cell layer.

**Figure 3 ijms-24-15277-f003:**
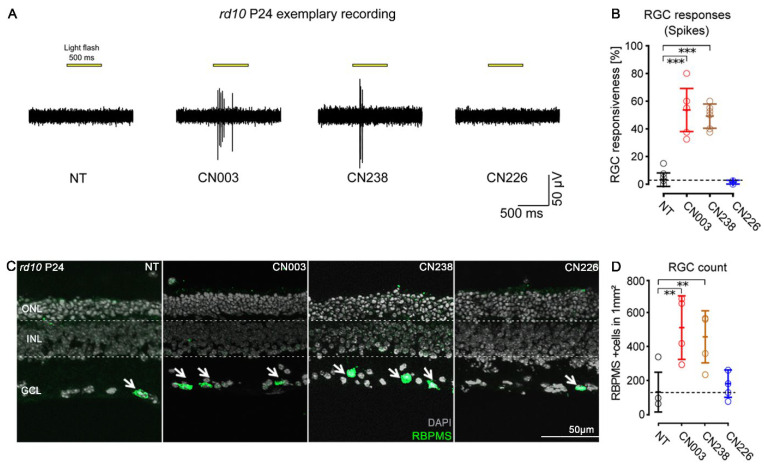
CN238 improves retinal ganglion cell viability and function in *rd10* retina. (**A**) Representative light-correlated retinal ganglion cell (RGC) spike trains recorded using microelectrode array (MEA) on P24 *rd10* retinal explants. Retinae were treated from post-natal day (P) 14 to P24 with 50 µM of CN003, CN238, or CN226, and compared to non-treated (NT) retinal explants. (**B**) Quantification of light-evoked *rd10* RGC activity in NT and treated explants (% MEA electrodes detecting light-correlated spike-activity). *n* ≥ 4 retinae from different animals. (**C**) Sections derived from recorded *rd10* retinal explant cultures were stained with DAPI (grey) and RBPMS (green). Arrows point to RGCs stained with RBPMS. (**D**) Quantification of RBPMS-positive cells in *rd10* P24 retinal explant sections. *n* ≥ 4 retinae from different animals. Error bars indicate SD; statistical analysis in B, D: one-way ANOVA followed by Dunnett’s multiple comparison test; significance levels: ** *p* ≤ 0.01, *** *p* ≤ 0.001. ONL = outer nuclear layer, INL = inner nuclear layer, GCL = ganglion cell layer.

**Figure 4 ijms-24-15277-f004:**
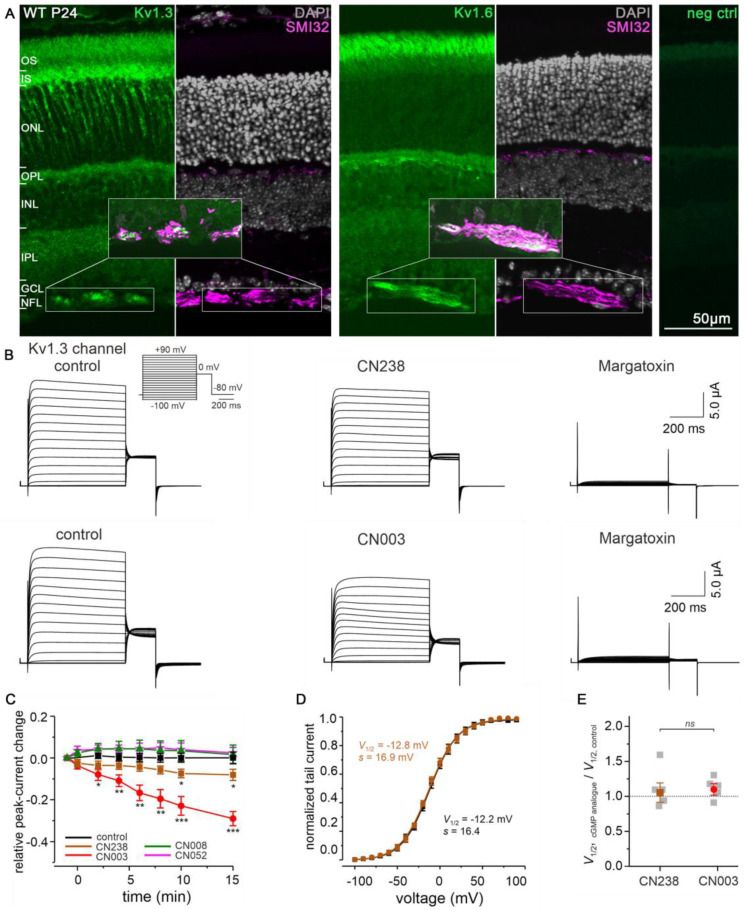
Retinal localization of Kv1.3 and Kv1.6, and effects of PKG inhibitors on Kv1.3 channels. (**A**) Retinal cross-sections derived from post-natal day−24 (P24) wild-type (WT) mice stained with DAPI (grey), for Kv1.3, Kv1.6 (both green), or the axonal marker SMI32 (magenta). (**B**) Representative whole-cell recordings showing effects of 50 µM CN238 (upper panel), 50 µM CN003 (lower panel), and 500 nM margatoxin on Kv1.3 channels expressed in *Xenopus laevis* oocytes. Voltage protocol depicted at top left (cf. Methods). (**C**) Relative Kv1.3 current change after application of cGMP analogues to the bath solution. Data points represent means (± SEM) from 4–7 measurements of the peak currents measured at +60 mV. The maximal effect was observed after 15 min incubation (* *p* ≤ 0.05, ** *p* ≤ 0.01, *** *p* ≤ 0.001, cf. Appendix A). (**D**) Plot showing normalized tail currents at 0 mV as a function of the depolarizing test voltages, recorded in absence (black symbols) or in presence of CN238 (brown symbols). Data points represent means (± SEM) from seven measurements. The curves are best fits with Equation (1). (**E**) Comparison between the *V*_1/2_ values obtained after application of either CN238 (brown, *n* = 7) or CN003 (red, *n* = 7). Gray symbols represent individual measurements. No significant difference was observed in Kv1.3 voltage dependence under all experimental conditions. OS = outer segment, IS = inner segment, ONL = outer nuclear layer, OPL = outer plexiform layer, INL = inner nuclear layer, IPL = inner plexiform layer, GCL = ganglion cell layer, NFL = nerve fibre layer.

**Figure 5 ijms-24-15277-f005:**
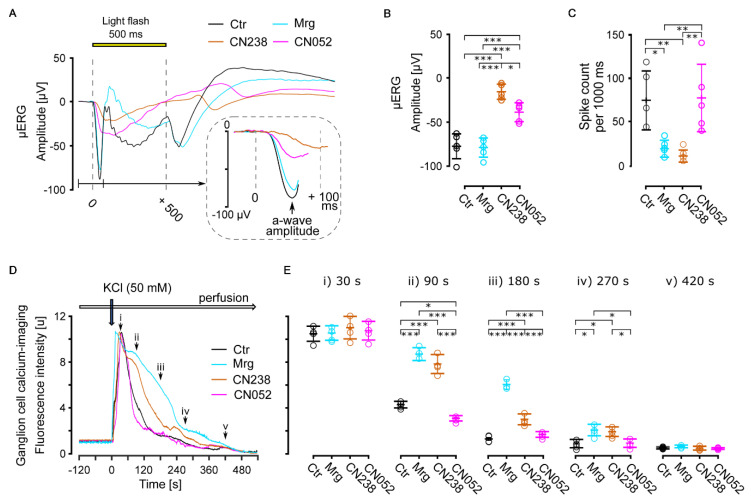
Effects of PKG-modulators on retinal photoreceptors and ganglion cells. (**A**–**C**) Multi-electrode array recordings of light-stimulation-induced retinal responses (wild-type mice, *n* = 5 retinae) under control conditions and in presence of margatoxin (Mrg, 50 nM), CN238 (50 µM), or CN052 (50 µM). Representative micro-electroretinogram (µERG) traces (**A**), µERG a-wave amplitude, (**B**) and retinal ganglion cell (RGC) spike counts (**C**). (**D**,**E**) Ca^2+^−imaging recording of KCl-induced RGC responses (adult *rd10* mice, *n* = 4 retinae, 200 cells per recording) under control conditions and in presence of Mrg, CN238, or CN052 (concentrations as in (**A**–**C**)). Representative KCl-evoked (50 mM) RGC traces (**D**) and values of intracellular Ca^2+^ clearance at 30, 90, 180, 270, and 420 s post stimulation (**E**). Error bars indicate SD; statistical analysis: one-way ANOVA followed by Dunnett’s multiple comparison test; significance levels: * *p* ≤ 0.05, ** *p* ≤ 0.01, *** *p* ≤ 0.001.

**Figure 6 ijms-24-15277-f006:**
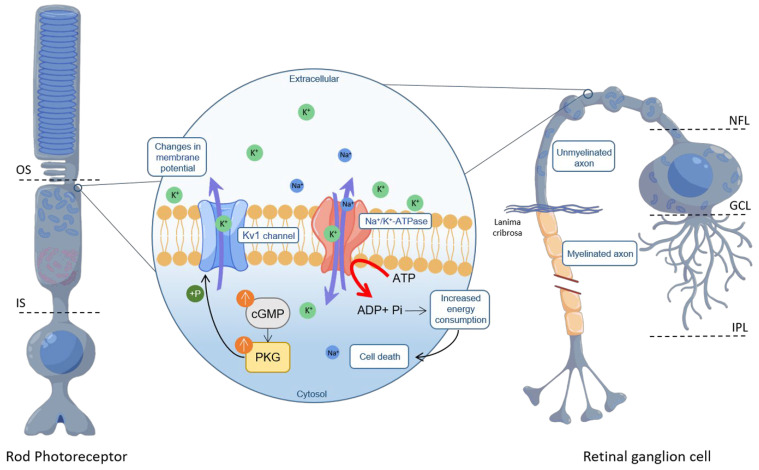
PKG-mediated Kv1 channel phosphorylation in photoreceptors and retinal ganglion cells. Phosphorylation of Kv1 channels because of excessive cGMP/PKG signalling can increase K^+^ outflow. This activates Na^+^/K^+^-ATPase in photoreceptor inner segments (IS) and in unmyelinated axon tracts of retinal ganglion cells (RGCs). High ATP consumption likely precipitates cell death. OS = outer segment, NFL = nerve fibre layer, GCL = ganglion cell layer, IPL = inner plexiform layer.

## Data Availability

All data generated or analysed during this study are included in this published article and its Appendix A.

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
