# Peer review of "The PKG Inhibitor CN238 Affords Functional Protection of Photoreceptors and Ganglion Cells against Retinal Degeneration"

_ijms, 2023, doi:10.3390/ijms242015277_

Round 1

Reviewer 1 Report

this is a very interesting article 

Author Response

We thank the reviewer for his comment. 

Reviewer 2 Report

In this manuscript, the author Tolone et al, investigate the potential protective effect of a new PKG inhibitor (CN238 is a cGMP) in hereditary retinal degeneration by investigating the beneficial effects on photoreceptor and retinal ganglional cells using two common mice models (rd1 and rd10 mutant). Collectively, the results show that important neuroprotective effects including preservation of photoreceptor and ganglion cell viability. This is a good paper, well written with appropriate method and strong and convincing results.

General minor comments:

1-vc003 and cn004 show similar strong effects on fig1a apoptosis, and strong similar effects on ganglion cell viability for cn003, how does cn238 differ from others; clarify this in more detail in the discussion.

2-Although the molecular mechanism has not been investigated, by what downstream molecular mechanism does PKG inhibition affect photoreceptor and ganglion cell survival? In addition to the tunel, a microarray on the level of apoptotic factor in the retina will be suggested.

3-The number of animals (N) should be added in the figure legends for each experiment.

4-Could CN238 have beneficial effects on retinal and choroidal vasculature; please comment in the discussion?

Author Response

Reviewer 2

In this manuscript, the author Tolone et al, investigate the potential protective effect of a new PKG inhibitor (CN238 is a cGMP) in hereditary retinal degeneration by investigating the beneficial effects on photoreceptor and retinal ganglional cells using two common mice models (rd1 and rd10 mutant). Collectively, the results show that important neuroprotective effects including preservation of photoreceptor and ganglion cell viability. This is a good paper, well written with appropriate method and strong and convincing results.

General minor comments:

  1. cn003 and cn004 show similar strong effects on fig1a apoptosis, and strong similar effects on ganglion cell viability for cn003, how does cn238 differ from others; clarify this in more detail in the discussion.

Response: We thank the reviewer for this suggestion. Like the reference retinoprotective compounds CN003 and CN004, CN238 has a cGMPS backbone containing a sulphur-modified phosphate in Rp configuration, which confers PKG inhibitory properties (cf. Figure S1). Thus, CN238 is structurally similar to CN003 and CN004. Yet, especially the results on the functional protection suggest that CN238 possesses superior protective abilities when compared to the previous reference compounds. The superior efficacy of CN238 is reported in the results section (page 4, lines 166-171). In the discussion this finding was mentioned on page 12, lines 387-388. To highlight this point further we have inserted another corresponding sentence at the very beginning of the discussion (page 11, lines 371-372) and at the end (page 15, lines 532-533).

  1. Although the molecular mechanism has not been investigated, by what downstream molecular mechanism does PKG inhibition affect photoreceptor and ganglion cell survival? In addition to the tunel, a microarray on the level of apoptotic factor in the retina will be suggested.

Response: This is indeed an important question, which unfortunately we can on only partially address since the downstream pathways of PKG in degenerating photoreceptors and retinal ganglion cells are yet poorly understood. Increased PKG signalling in photoreceptors has been linked to increased activity of poly-ADP-ribose-polymerase (PARP), histone deacetylase (HDAC), and calpain proteases, all known to be implicated in photoreceptor cell death (Arango-Gonzalez et al., 2014). Intracellular changes of known PKG targets, such as vasodilator-stimulated phosphoprotein (VASP) and cAMP response element binding protein (CREB), have been seen in dying photoreceptors (Paquet-Durand et al., 2009). We have tried to address this problem in a recent study, where we used multiplex peptide microarrays to measure PKG1- and PKG2-mediated phosphorylation (Roy et al., 2022). This study identified several novel PKG substrates potentially related to RD and validated their retinal expression in murine tissue. Among the substrates discovered was also Kv1.3, on which we focused in the present study. As far as the degenerative mechanisms upstream of PKG are concerned, we discuss the possibility that the mutation-dependent rise of cGMP in photoreceptors, could be mirrored by a nitric oxide -induced cGMP production in retinal ganglion cells (page 13, lines 451-461).

  1. The number of animals (N) should be added in the figure legends for each experiment.

Response: We thank the reviewer for spotting this. We have now added the respective number of animals to each figure legend.

  1. Could CN238 have beneficial effects on retinal and choroidal vasculature; please comment in the discussion?

Response: This is another interesting question. The cGMP-PKG pathway is known to regulate vessel relaxation and thereby blood pressure. Since in the present study we used retinal explants devoid of vasculature, the neuroprotective effects observed are independent of vascular effects. Whether additional vascular effects could play a role in the in vivo situation is an open question, however, we note that with previous in vivo studies using the parent compound CN003, we did not see changes in blood pressure in mice, rats, or macaque monkeys (unpublished observations). To address this question from the reviewer, we have now added an additional paragraph to the discussion (page 13, lines 461-470).

Reviewer 3 Report

Thanks the authors for conducting this research and writing up this basic science biochemistry animal study results.

Similar manuscript by the same group of authors was posted on preprint server biorxiv, since 2021. And was cited at least 5 times, by even review articles as well.

Author Response

We thank the reviewer for his comments.  

Reviewer 4 Report

The authors explore the therapeutic effects of inhibition of cGMP-dependent protein kinase G on photoreceptor cell loss and retinal ganglion cell viability in two different retinal degeneration animal models. The manuscript resulted from a detailed and well-conceived study with a clearly defined hypothesis testing mechanisms of action, a clear premise based on published data, and adequate controls, both scientific and technical. The inclusion of functional analyses assessing retina health as well as meaningful statistical analyses and carefully drawn conclusions add to the quality of the study.

Minor concerns relate to the presentation and description of data / data analysis and should be addressed, but don’t distract from the high quality of the manuscript:

·       Results from the statistical assessments of data quantitation should be discussed in more detail, specifically keeping in mind non-linear and threshold effects.

·       When pseudo-coloring immunostaining the color indicating colocalization should be different from the color used for other labels (e.g. Fig. 4A).

·       Adding quantitative measures for colocalization should be considered.

·       The number and nature of replicates and a description of specific statistical methods used should be expanded or, in some cases, added to the figures and results section.

·       Minor language editing is needed. E.g., the term “exemplary” should be replaced with “representative” when describing representative data.

Minor language editing is needed. E.g., the term “exemplary” should be replaced with “representative” when describing representative data.

Author Response

Reviewer 4

The authors explore the therapeutic effects of inhibition of cGMP-dependent protein kinase G on photoreceptor cell loss and retinal ganglion cell viability in two different retinal degeneration animal models. The manuscript resulted from a detailed and well-conceived study with a clearly defined hypothesis testing mechanisms of action, a clear premise based on published data, and adequate controls, both scientific and technical. The inclusion of functional analyses assessing retina health as well as meaningful statistical analyses and carefully drawn conclusions add to the quality of the study.

Minor concerns relate to the presentation and description of data / data analysis and should be addressed, but don’t distract from the high quality of the manuscript:

  1. Results from the statistical assessments of data quantitation should be discussed in more detail, specifically keeping in mind non-linear and threshold effects.

Response: We thank the reviewer for alerting us to this topic. We would like to note that the statistical analysis used here is in accordance with the scientific standards used for similar such studies. Nevertheless, in terms of non-linear and threshold effects an interesting observation may relate to the comparison of neuroprotective effects of CN003 vs. CN238 in Figures 1 and 2: In the comparative cell death / cell survival analysis shown in Figure 1B and 2B, CN238 appears to perform only marginally better, if better at all, than CN003. However, in the functional analysis shown in Figures 2D and its statistical analysis in Figure 2E, CN238 clearly and highly significantly outperforms CN003. This outcome may indeed be related to threshold effects where a marginal increase in the overall viability of the photoreceptor cell population strongly and non-linearly improves the function of that population. We have now inserted a corresponding comment in the results section for Figure 2E (page 4, lines 166-167).

  1. When pseudo-coloring immunostaining the color indicating colocalization should be different from the color used for other labels (e.g. Fig. 4A).

Response: We have reviewed this and adjusted the coloring so that colocalization in Figure 4A is more evident. We note that colocalization in high resolution micrographs depends on the exact subcellular localization of the given proteins. Thus, while for instance SMI32 and Kv1.3 or Kv1.6 may very well be localized to the same ganglion cell axon, a slight offset in axonal localization can produce different coloring.

  1. Adding quantitative measures for colocalization should be considered.

Response: Please refer also to the previous comment. We have indeed considered this, however, since we cannot be sure that the axonal marker used here (SMI32) really has the same subcellular localization on an individual ganglion cells axon as our target proteins (Kv1.3, Kv1.6), we feel that quantitation of overlap may produce misleading results / conclusions. Moreover, given the fact that there are at least 40+ different types of ganglion cells, not all of which may display SMI32 or Kv1.3 / Kv1.6 expression, the significance of quantified colocalization data would be limited at best.

  1. The number and nature of replicates and a description of specific statistical methods used should be expanded or, in some cases, added to the figures and results section.

Response: We thank the reviewer for spotting this. We have now added the respective number of animals to each figure legend. The specific statistical methods used are explained in the supplementary information, as referred to in the main text (line 743 - 744)

  1. Minor language editing is needed. E.g., the term “exemplary” should be replaced with “representative” when describing representative data.

Response: We thank the reviewer for noticing this. We have changed the word 'exemplary' to 'representative'.

Comments on the Quality of English Language

Minor language editing is needed. E.g., the term “exemplary” should be replaced with “representative” when describing representative data.

Reviewer 5 Report

Please see detail report attached.

Overall, well designed and conducted research and well written article. However, I recommend moderate (but feasible) revision before accepting it for publishing!

Good Luck!

Author Response

We thank the reviewer for his comments. 

Round 2

Reviewer 5 Report

I don't see recommended changes were made or considered, and still could Not find Supplementary data file!

Author Response

Reviewer 5

In the presented study, authors have addressed an important issue of developing potential a mutation-independent therapy for a heterogeneous retinal degenerative diseases resulting in blindness. They have identified a novel chemical compound named CN238 and demonstrated its effectiveness in preserving photoreceptors and retinal ganglion cells in retinal explants from two different mice models rd1 and rd10. The data presented support the claim of cellular preservation in rd10 retina explants but lacks data for functional preservation in the diseased retinal explants! Furthermore, authors have commendably discussed how the presented study should be verified in vivo, and potential challenges in drug administration! Overall, well designed and conducted research and well written article. However, I recommend moderate (but feasible) revision before accepting it for publishing!

Major comments:

  1. Could Not find Supplementary data file!

Response: We have provided a supplementary information document, together with the main manuscript. We believe this should now be available to the reviewers. Please do not hesitate to contact IJMS, should the supplementary information document somehow not be accessible.

2.2. PKG inhibition preserves viability and function of photoreceptors in rd10 retinal explants.

  1. WT control for MEA missing, to clarify any possible noise/effect of the drugs CNs. This is clarified in Line 237-238: “The light-induced hyperpolarization response in the μERG obtained from P24 WT explants did not differ between NT and CN238-treated retinae…”, however Need to see data from Supplementary information.

Response: We thank the reviewer for this point. The WT MEA results are indeed shown in Figure S3 of the supplementary information. The supplementary information should be available together with the main manuscript (see above).

2.3. CN238 prevents axotomy-induced degeneration of retinal ganglion cells

  1. In lines 207-209; it should be clarified that the % represented are Not of RGC numbers, rather they the % MEA electrodes detecting light-correlated spike-activity (As described in figure legend).

Response: We thank the reviewer for this comment. We have added a sentence (page 6, lines 207-208) to clarify that the “light responsiveness of RGCs” (Figure 3B) corresponds to the percentage of MEA electrodes detecting light-related spiking activity.

  1. Did CN238 showed similar or at least some level protective effect on rd1 retinal explants? If yes, then how similar, or different was it compared to the rd10 explants?

Response: We thank the reviewer for this comment. Indeed, we did not evaluate the effects of CN238 on RGCs degeneration in rd1. However, results obtained on WT long-term retinal explant cultures suggest that the loss of RGCs is a mutation-independent phenomenon and takes place because of optic nerve transection (as described in the main manuscript on page 7 in lines 240-259 and shown in Figure S3).

2.4. Effects of cGMP-mediated inhibition of PKG in retinal ganglion cells

  1. What was the level of phosphorylated potassium channels in untreated rd10 retinal explants? Were they high (compared to WT) as in case of rd1 retinal explants?
  2. Upon treatment, how much (%) of the decreased phosphorylation level compared to the treated WT retinal explants?

Response: Unfortunately, we do not have phosphorylation data comparing WT to rd10. However, we did compare untreated rd10 vs. CN238 treated rd10 retinal explants. Here, CN238 treatment produced a numerically small but statistically significant reduction in Kv1.6 (KCNA6) phosphorylation (arbitrary units; untreated rd10: 14.778 ± 0.1 SD; CN238 rd10: 14.625 ± 0.11 SD; n = 9, p < 0.01). We note that the fluorescence-based, highly non-linear phosphorylation signal intensity obtained in the protein phosphorylation array screening does not allow to directly deduce the quantity of phosphorylated / non-phosphorylated channels. Given the fact that the signal differences obtained are rather small (approx. 1% reduction) we feel that reporting this result here – even though it supports our conclusions – would be misleading.

2.5. Down-stream effectors of CN238 include Kv1 channels.

  1. How different (quantitatively & localization) are the 3 and Kv1.6 channels in rd1 (at P18) & rd10 (at P24) compared to the WT?

Response: We thank the reviewer for this question. We have now included in the supplementary information a new figure, Figure S9 depicting the localisation of Kv1.3 and Kv1.6 in wild-type P11, rd1 P11 and rd10 P24 retinas. In both WT and rd1 P11 Kv1.3 was localised in the ONL, probably in photoreceptor axons, and in two discrete sublaminae of the IPL, where the synapses between bipolar cell axons and ganglion cell dendrites reside. In rd10 P24, Kv1.3 was localised in the nerve fibre layer (NFL). Kv1.6 was prominently expressed in the ganglion cell layer (GCL) and nerve fibre layer (NFL) in all three investigated mouse models.

  1. As demonstrated previously with the drug CN003, authors should also provide direct evidence of the new drug CN238 being PKG inhibitor. Or there aren’t any potassium channels remained preserved in rd-retinae?

Response: We thank the reviewer for this question. Like the reference retinoprotective compound CN003, CN238 has a cGMPS backbone containing a sulphur-modified phosphate in Rp configuration, which confers PKG inhibitory properties (Figure S1). Furthermore, the inhibitory potential of CN238 was tested on recombinant human PKG isoforms PKG1α, PKG1β, and PKG2. This data is shown in Figure 1C and indicates that CN238 is a somewhat more effective PKG inhibitor, notably for PKG1β and PKG2.

2.6. CN238 inhibits Kv1-mediated Ca2+ extrusion.

  1. What are the effects in rd1 & rd10 retina explants? What level of functional preservation achieved with CN238 in those diseased retinae?

Response: In rd1 retinal explants CN238 reduced the percentage of dying, TUNEL positive cells as shown in Figure 1B. The level of functional preservation achieved with the CN238 in rd10 retinal explants is presented in Figure 2D, E, and F. Furthermore, the functional preservation of WT RGCs is shown in Figure S3.

For rd1 retinal explants µERG functional recordings were not performed as rd1 photoreceptors degenerate before P18, i.e. before synaptic connectivity between photoreceptors and bipolar cells and bipolar cells and RGCs are formed and matured. Thus, even if a significant number of rd1 photoreceptors could be preserved by CN238 until P24, with the methodology at hand a functional benefit will be difficult, if not impossible, to demonstrate in rd1 retina.

Minor comments

  1. PKG1a_activation by CN226 line connecting solid blue dots missing in figure 1C
  2. Scale bar-line is missing/not visible in figure 1A
  3. Missing “n” value for figure 3B & 3D

Response: We thank the reviewer for noticing these mistakes. We have now corrected Figure 1 and included information about the number of biological replicates in the legend to Figure 3.